# Physicochemical and Sensory Evaluation of Sustainable Plant-Based Homopolymers as an Alternative to Traditional Emollients in Topical Emulsions

**DOI:** 10.3390/pharmaceutics17020265

**Published:** 2025-02-17

**Authors:** Talita Ganem Meneguello, Nathalia Kopke Palma, Yasmin Rosa Santos, Ariel Figueira Carvalho, Ariane Dalan da Silva Ladeira, Fabiana Perrechil Bonsanto, Newton Andreo-Filho, Patricia Santos Lopes, Heather Ann Elizabeth Benson, Vania Rodrigues Leite-Silva

**Affiliations:** 1Programa de Pós-Graduação em Medicina Translacional, Universidade Federal de São Paulo, Rua Pedro de Toledo, 720, São Paulo 04039-002, Brazil; talita.meneguello@unifesp.br (T.G.M.); yasmin.rosa@unifesp.br (Y.R.S.); 2Departamento de Ciências Farmacêuticas, Instituto de Ciências Ambientais, Químicas e Farmacêuticas, Universidade Federal de São Paulo, Rua São Nicolau, 210, Diadema 09913-030, Brazil; nkpalma@unifesp.br (N.K.P.); ariel.carvalho@unifesp.br (A.F.C.); ariane.ladeira@unifesp.br (A.D.d.S.L.); newton.andreo@unifesp.br (N.A.-F.); patricia.lopes@unifesp.br (P.S.L.); 3Departamento de Engenharia Química, Universidade Federal de São Paulo, Rua São Nicolau, 210, Diadema 09913-030, Brazil; fabiana.perrechil@unifesp.br; 4Curtin Medical School, Curtin University, Perth, WA 6845, Australia; h.benson@curtin.edu.au; 5Frazer Institute, Faculty of Medicine, The University of Queensland, Brisbane, QLD 4102, Australia

**Keywords:** emollient, rheology, sensory analysis, neurosensory analysis, eye tracking, sustainability

## Abstract

**Objectives**: This study evaluated the potential of sustainably sourced, plant-based homopolymers derived from citronellol as an alternative to the traditional emollients used in pharmaceutical, cosmetic, and personal care products. With increasing emphasis on environmentally friendly ingredients and manufacturing processes, this study assessed the efficacy of these homopolymers in semi-solid and emulsion-based formulations. **Methods**: The analyses focused on physicochemical, sensory, biophysical, and neurosensory characteristics. **Results**: The results demonstrated that emulsions containing sustainable homopolymers maintained viscoelastic stability, preserving rheological properties over time under varying conditions. These formulations showed comparable structural and functional stability to those with traditional emollients while offering skin hydration, moisture retention, and elasticity, with reduced transepidermal water loss. Sensory evaluations highlighted positive user acceptance, with participants favoring the skin feel and in-use qualities of these emulsions over synthetic alternatives. Neurosensory analyses confirmed the strong visual appeal of the product packaging, capturing user attention effectively. **Conclusions**: These findings underline the capability of plant-based homopolymers to replace traditional emollients while providing significant consumer appeal and sustainability benefits. This study establishes their potential as viable components in the development of more eco-friendly topical formulations for the pharmaceutical, cosmetic, and personal care industries.

## 1. Introduction

Emollients are key ingredients that provide moisturization to the skin and contribute to the sensory qualities of smoothness, softness, and shine in topical pharmaceutical and cosmetic products. Traditional ingredients commonly used in topical formulations have long been recognized for their efficacy in achieving desired therapeutic or cosmetic effects. However, a critical evaluation of their sourcing and processing methodologies reveals significant challenges. For instance, lanolin, derived from sheep’s wool, and palm oil derivatives are often associated with ethical and environmental concerns, including biodiversity loss, unsustainable agricultural practices, and significant carbon emissions. Similarly, mineral-based ingredients, such as mineral oil, while functionally effective, are typically extracted through mining operations that contribute to environmental degradation and may involve the socio-economic exploitation of local communities. These issues raise fundamental ethical and ecological concerns, emphasizing the need to develop sustainable and ethically sourced alternatives [1].

To meet the growing demand for sustainable, natural, and ethical products, considerable research efforts are being directed toward identifying natural ingredients and responsible processing methods that satisfy the dual criteria of environmental sustainability and social responsibility while also delivering high-quality, effective products [2].

An example of innovation in this area is the incorporation of phytosphingosine, a lipid derived from yeast fermentation, as an alternative to ceramides. Simpson et al. demonstrated that phytosphingosine exhibits superior skin barrier restoration and anti-inflammatory properties and is highly effective in managing atopic dermatitis and other skin conditions [3]. Another example is the synthesis of biopolymeric chitosan nanoparticles derived from shrimp shells, which serve as a replacement for conventional preservatives. These nanoparticles exhibit enhanced antimicrobial activity and lower cytotoxicity compared to traditional chemical preservatives such as parabens, thereby improving product safety and shelf life [4].

A third example involves the use of biosurfactants, such as sophorolipids, which exhibit comparable or superior emulsifying and cleansing properties while being biodegradable and less irritating to the skin. These attributes align with the growing demand for environmentally friendly and dermo-compatible products [5]. These examples highlight the potential of innovative ingredients to enhance performance and safety profiles while addressing sustainability and ethical considerations.

Natural emollients are substances derived from botanical or animal sources that provide hydration, enhance skin barrier function, and improve the overall texture and appearance of the skin. Common examples include shea butter, jojoba oil, and almond oil. These ingredients are rich in essential fatty acids, vitamins, and antioxidants, making them effective in restoring skin moisture, reducing transepidermal water loss (TEWL), and soothing irritated or sensitive skin [6]. Unlike synthetic counterparts, natural emollients often offer enhanced biocompatibility and minimal risk of adverse reactions, which makes them highly desirable in formulations for sensitive or dry skin.

In recent years, plant-based polymers have emerged as sustainable alternatives to traditional emollients. Derived from renewable resources such as cellulose, alginate, and starch derivatives, these polymers not only provide emollient properties but also contribute to the stability, viscosity, and texture of topical formulations. Their ability to form biocompatible and biodegradable films on the skin offers long-lasting hydration and a smooth sensory profile [7]. This development aligns with the increasing demand for eco-friendly and skin-compatible products in the cosmetic and pharmaceutical industries.

Policitronellol is a polymer derived from citronellol, a monoterpenoid alcohol found in essential oils such as citronella, rose, and geranium oils. Homopolymers of policitronellol are synthesized by polymerizing citronellol into long chains, resulting in a material with unique physicochemical properties. These polymers exhibit excellent film-forming capabilities, high hydrophobicity, and low water solubility, making them potential candidates for use as emollients in topical formulations [8,9].

The interest in policitronellol stems from its natural origin and its potential to replace petroleum-based materials. Early studies suggest that homopolymers of policitronellol offer superior occlusive properties, enhancing moisture retention on the skin while providing a non-greasy, lightweight texture. Additionally, policitronellol’s biodegradability and compatibility with other natural ingredients make it a sustainable choice for modern formulations [9].

Studies highlight the potential of policitronellol homopolymers in addressing environmental and ethical concerns while maintaining performance. For instance, research has shown that these polymers enhance moisture retention and provide a protective barrier without compromising skin comfort [9].

The decision to investigate policitronellol homopolymers in this study arises from their promising combination of properties; they are naturally derived, possess strong emollient capabilities, and align with sustainability principles. Their ability to provide hydration and barrier protection, coupled with a pleasant sensory profile, positions them as ideal candidates for next-generation topical products. Further research into their molecular structure and interactions in formulations could unlock new applications in skincare and pharmaceuticals.

This study evaluated the physicochemical properties of emulsions formulated with policitronellol homopolymers, including their rheological properties. In vivo studies with human volunteers were conducted to determine their hydration efficacy compared with traditional emollients. Additionally, as emollients are the key ingredients that provide the sensory properties of topical products when applied to the skin, the effects of these plant-based emollients were compared to traditional emollients in a study analyzing the efficacy of emulsions by examining their physicochemical, sensory, biophysical, and neurosensory characteristics.

## 2. Materials and Methods

This study was approved by the Research Ethics Committee (CEP No.: 0086/2023) of the Federal University of São Paulo (UNIFESP). All participants were properly informed about the experiment and signed the Informed Consent Form (ICF) before the experiment started.

### 2.1. Materials

The raw materials used for the formulation in this work were: Cetearyl Alcohol (Aqia, Guarulhos, SP, Brazil), Glyceryl Stearate (Aqia, Guarulhos, SP, Brazil), Ceteareth-20 (Aqia, Guarulhos, SP, Brazil), Sorbitan Stearate (Aqia, Guarulhos, SP, Brazil), Polycitronellol HA (P2Science, Woodbridge, NJ, USA), Dimethicone 100 (Labsynth, Diadema, SP, Brazil), Polycitronellol H (P2Science, Woodbridge, NJ, USA), Dimethicone 350 (Labsynth, Diadema, SP, Brazil), Polycitronellol and Euphorbia Cerifera (Candelilla) wax (P2Science, Woodbridge, NJ, USA), Petrolatum (Labynth, Diadema, SP, Brazil), and Phenoxyethanol (Proserv, São Paulo, SP, Brazil).

### 2.2. Preparation of Emulsions

Seven emulsions were developed, one without emollients and six with emollients, as outlined in (Table 1).

The base formulation (A), without the addition of emollients, was prepared by heating the raw materials of the oil phase to 75 °C for 15 min, and was followed by the addition of the aqueous phase, which was also heated to 75 °C for 10 min under mechanical stirring (715 Fisatom) at 1000 rpm. The mixture was then cooled under stirring, with the preservative Proteg PF IV added at 40 °C. Formulations B, C, D, E, F, and G followed the same procedure, with the addition of different emollient components to the oil phase, as outlined in (Table 1).

### 2.3. Rheological Analysis

The rheological properties of the emulsion formulations were analyzed using an MCR 92 rheometer (Anton Paar, Graz, Austria) equipped with a 25 mm diameter parallel plate geometry and a 0.2 mm gap. Flow curves were generated from an up-down-up step program with the shear rate ranging from 0 to 300 s^−1^. Thixotropy was estimated by the hysteresis area between the first and the third curves. The second curve was used to obtain the rheological parameters by fitting the Herschel–Bulkley model (Equation (1)):(1)$$\sigma =\sigma_{0}+k\cdot {\dot{\gamma }}^{n}$$
where σ is the shear stress (Pa), σ0 is the yield stress (Pa), k is the consistency index (Pa.s^n^), n is the flow behavior index (dimensionless), and $\dot{\gamma }$ is the shear rate (s^−1^).

All measurements were conducted in triplicate at a controlled temperature of 35 °C.

### 2.4. Sensory Analysis with a Trained Panel

The sensory evaluation of the formulations was conducted in sensory booths with 14 female evaluators, aged between 25 and 30 years, who were trained and validated as skin product panelists. The samples were prepared in a standardized manner, with the emulsions anonymized and identified by random three-digit codes to prevent bias. Each portion was measured in equal volumes (25 µL) and presented in appropriate containers, such as soap dishes, ensuring uniformity. During the test execution, the panelists applied each sample to standardized areas of the skin, such as the forearm, evaluating predefined sensory attributes, including texture, spreadability, and absorption. Applications were performed at defined 15-min intervals to prevent interference between samples, and each formulation was evaluated in triplicate, ensuring data reproducibility and reliability. The results were collected and subsequently analyzed statistically to identify significant differences between the formulations, resulting in a total of 42 responses. The statistical method used was ANOVA (source of variation: sample and evaluator) followed by Fisher’s Least Significant Difference (LSD) test, with statistical analyses of means and standard deviations for each attribute at the 5% significance level. The software utilized for analysis was XLSTAT 2023 (Lumivero, Denver, CO, USA).

### 2.5. Hydration and Transepidermal Water Loss Test

The moisturizing efficacy of the emulsions was evaluated by measuring skin hydration and water loss following the application of each emulsion. The water content in the stratum corneum was analyzed using a Corneometer^®^ CM 825 (Courage—Khazaka^®^, Cologne, Germany), and transepidermal water loss (TEWL) was determined using a Tewameter^®^ TM 300 (Courage—Khazaka^®^, Cologne, Germany). The formulations were applied (2 mg/cm^2^) in one single application randomly to four marked 9 cm^2^ squares on the forearms of 13 participants (aged between 18 and 60 years) in a climate-controlled environment under conditions of 20 °C ± 2 °C and relative humidity of 40% ± 5%. There were 10 Corneometer^®^ measurements per quadrant and a continuous 1-min measurement for TEWL. All measurements were performed at 1, 2, 3, and 4 h after the application of the formulations. Comparative statistics with paired data were used, adopting a 95% confidence level.

### 2.6. Neurosensory Analysis

This study involved 105 participants, both male and female, aged between 18 and 40 years, healthy, with no history of neurological diseases and not experiencing headaches, nausea, or vertigo at the time of the experiment. The eye-tracking instrument used was the Tobii^®^ Pro Fusion (Tobii AB, Stockholm, Sweden); this instrument contains infrared cameras to capture eye movements. Using the eye-tracking method, participants were instructed to sit in front of a computer screen where images would appear for observation and eye movement capture. To begin the test, device calibration was required, with a fixation cross appearing on the screen for standardization. After this process, participants viewed, for 5 s, one matrix at a time containing two packages, allowing for the capture of eye movements. Each timeline contained a sequence of images featuring green and white packaging labeled as “plant-based moisturizer” and “silicone-based moisturizer”, presented in a randomized order. To allow for eye rest between images, a screen with a plus sign (+) was displayed. Participants viewed the images freely, without moving their head or neck. Finally, a thank you screen appeared, following the sequence (Figure 1).

#### Consumer Profile and Product Preference

To analyze consumer preferences between plant-based and synthetic products, a study was conducted involving 105 healthy participants, both male and female, aged between 18 and 40 years.

Participants completed a questionnaire with questions that categorized them based on their responses, such as their purchasing preference for plant-based or synthetic products, or if they had no preference. Based on their answers, participants indicated whether they would pay 20% more for plant-based or synthetic products, explained why they preferred cosmetics formulated with plant-based or synthetic raw materials, and shared their thoughts on products made with these types of raw materials. If a participant had no preference, they did not complete this questionnaire. In the final question, participants were asked to select which statements they considered to be correct regarding sustainability, concluding their participation.

## 3. Results

### 3.1. Rheological Analysis

The analysis of the rheological curves showed that most formulations exhibited thixotropic behavior, in which the viscosity decreases with time, showing a hysteresis loop on the rheogram (a plot of shear stress vs. shear rate) between up-down-up shear rate ramps. The area enclosed by this loop represents the energy required to break down the internal structure of the sample during shear and it can be associated with more structured systems. After removing the thixotropy, the samples showed pseudoplastic behavior, in which the apparent viscosity decreases with the increase in shear rate (Figure 2).

Table 2 shows the rheological parameters of the formulations. Yield stress (σ0 in Pa) represents the minimum force required for the material to start flowing. This parameter is crucial for materials such as pastes and gels, which remain solid until a sufficient force is applied. The consistency index (k, in Pa.s^n^) indicates the material’s resistance to flow, particularly at low shear rates. A higher k value reflects greater viscosity and flow resistance. Lastly, the apparent viscosity at 100 s^−1^ (η_100s_^−1^, in mPa.s) is the viscosity measured at a specific shear rate, providing a useful reference point for practical use or processing conditions. These parameters, when combined, offer a comprehensive view of the rheological behavior of materials, helping to predict their performance in different applications, such as stability during storage, ease of application, and behavior under external forces.

Pseudoplasticity can be confirmed by the flow behavior indexes lower than 1. Formulations A (base) to F (polycitronellol compound) had n values between 0.44 and 0.51, indicating a more pronounced pseudoplastic behavior. In contrast, formulation G (solid petrolatum) exhibited lower pseudoplasticity, with n closer to 1 (Table 2).

For the formulations analyzed, Formulations B (polycitronellol HA), D (polycitronellol H), and G (solid petrolatum) demonstrated the highest thixotropy values, indicating that these materials possess highly reconfigurable internal networks and dissipate more energy when subjected to shear cycles. In contrast, Formulation C (dimethicone 100), with a reported thixotropy value of zero, indicates no significant difference between the upward and downward curves in the rheogram. This suggests that the material has unstructured behavior or does not undergo restructuring after shear stress is applied.

Moreover, Formulations A (base), E (dimethicone 350), and F (polycitronellol compound) exhibited the highest viscosities (~1400 mPa.s at 100 s^−1^), while Formulations C (dimethicone 100), B (polycitronellol HA), and D (polycitronellol H) showed viscosities between 1000 and 1100 mPa.s. Formulation G (solid petrolatum) had the lowest viscosity at steady state, approximately 100 times lower than the others.

### 3.2. Sensory Analysis

Sensory evaluation is crucial for the acceptance or rejection of topical formulations, and methods that assess sensory perceptions are essential to ensure a successful product with good “in-use” characteristics and end-user experience. The results presented in Table 3, Table 4 and Table 5 include F-test values and *p*-values for each sensory attribute evaluated, identifying significant differences between formulations. These results are complemented by Levene’s test to verify the homogeneity of variances, ensuring the reliability of the analyses.

The data presented in Table 3 indicate significant differences among the formulations in the attributes of velvety film, dry touch, and residual greasy film. Formulation B (polycitronellol HA) demonstrated the highest levels of dry touch and velvety film compared to Formulation C (dimethicone 100), suggesting that polycitronellol HA can enhance these sensory attributes. Conversely, Formulation C exhibited an increase in residual greasiness. For all other evaluated attributes, including spreadability, glide, stickiness, white residue, shine, greasy film, oiliness, and absorption point, no significant differences were observed among the formulations, indicating consistency across these parameters.

When comparing Formulation D (polycitronellol H) with Formulation E (dimethicone 350), we observed no significant differences in most attributes: spreadability, glide, stickiness, white residue, residual white residue, shine, greasy film, oiliness, and absorption point. However, for stickiness, Formulation D received higher scores, suggesting greater tackiness compared to Formulation A (base) and Formulation E.

Regarding velvety film and dry touch, both Formulation D and Formulation E scored lower than Formulation A, indicating that the base without emollients provides a better velvety feel and a drier touch. While immediate shine showed no significant difference, residual shine was higher for Formulation D. The same trend was observed for oiliness and greasy film; no immediate differences were noted upon application, but the residual effect was more noticeable in Formulations D and E compared to the base. For the absorption point, Formulation E had the lowest score.

The data in Table 5 show significant differences among the formulations in the attributes of stickiness, velvety film, dry touch, residual shine, residual greasy film, and absorption point. Formulation G (solid petrolatum) exhibited higher residual shine and residual greasy film, while Formulation F (polycitronellol compound) stood out for its lower intensity of the velvety film. On the other hand, attributes such as spreadability, glide, immediate and residual white residue, immediate and residual oiliness, and immediate shine showed no significant differences among the formulations, indicating consistency in these parameters.

### 3.3. Evaluation of Hydration and Transepidermal Water Loss (TEWL)

This study compared the cosmetic formulations A, B, C; A, D, E; and A, F, G (Figure 3), assessing the average hydration levels at four different time points. Using paired comparisons with Bonferroni correction, statistically significant differences were identified between the following hydration means:

In the first comparison, formulations A (base), B (Polycitronellol HA), and C (Dimethicone 100) were used, based on previous analyses and sensory and physicochemical compatibility, to assess hydration using biophysical methods and in vivo testing with the Corneometer^®^.

This study evaluated the hydration levels provided by various cosmetic formulations over time. Formulation A (base), without polymers, showed a significant decrease in hydration after three hours, suggesting a dehydrating effect. In contrast, untreated skin showed an improvement in hydration, possibly influenced by external factors such as water intake or temperature, despite controlled variables.

Formulations B (polycitronellol HA) and C (dimethicone 100) maintained initial hydration levels, with Formulation B outperforming Formulation A during the first two hours, highlighting the effectiveness of sustainable polycitronellol HA compared to silicone. Formulation D (polycitronellol H) demonstrated superior hydration compared to Formulation E (dimethicone 350), with significant improvements observed at T1 and sustained through T3, while Formulation E presented lower hydration averages.

The moisturizing performance of Formulations F (polycitronellol compound) and G (solid petrolatum) was significantly better than Formulation A one hour after application (T1), with averages of 43.1 AU and 41.8 AU, respectively. Formulation F continued to improve hydration at T2 and T3, showing a gradual increase over time and indicating superior long-term moisturizing efficacy. These results highlight the potential of sustainable polycitronellol-based formulations as effective alternatives to traditional emollients.

### 3.4. Transepidermal Water Loss (TEWL) Measurements

This study analyzed the efficacy of different cosmetic formulations in maintaining skin hydration and reducing transepidermal water loss (TEWL) (Figure 4).

At the beginning of this study (T0), significant differences were observed between Formulations A (base), B (polycitronellol HA), and C (dimethicone 100) compared to the control. Formulations B and C exhibited similar average values and lower TEWL compared to the control, indicating higher efficacy in maintaining skin hydration.

One hour after application (T1), significant differences were observed between Formulation A and Formulations B and C (Figure 4B).

The TEWL means of Formulations A (base), D (polycitronellol H), and E (dimethicone 350) are presented in (Figure 4). A significant difference was observed only at the initial time (T0) (Figure 4A), before product application, between Formulation E (dimethicone 350) and the control. No other significant differences were noted among the other formulations at subsequent times. Formulations D (polycitronellol H) and E (dimethicone 350) showed similar average values one hour after product application, and both were lower than the control. At T2 (Figure 4C) and T3 (Figure 4D), the values for Formulation D increased to 13.4 UA and 14.3 UA, respectively, with T3 values exceeding those of the control. Therefore, Formulation E (dimethicone 350) demonstrated superior skin protection against transepidermal water loss (TEWL). As presented in Figure 4, the mean TEWL results for Formulations A, F, and G highlight significant differences between Formulation F (polycitronellol compound) and the control, as well as between Formulation G (solid petrolatum) and the control. Other formulations and times evaluated did not show significant differences.

Formulations F (polycitronellol compound) and G (solid petrolatum) demonstrated lower TEWL results than the control one hour after product application (T1), indicating effectiveness in preserving the skin barrier.

### 3.5. Neurosensory Analysis

Using eye-tracking technology to investigate consumer preferences for synthetic and plant-based cosmetic packaging, a paired *t*-test was employed to analyze the differences between the means. The results obtained are presented in (Figure 5).

Attention retention was significantly higher for the white packaging with a plant-based label (E4), with an average attention time of 3.89 s compared to the white packaging with a silicone label (E5), which had an average of 2.71 s. The E4 packaging achieved significantly higher average attention retention, reflecting a stronger preference or interest from consumers. According to the two-tailed test, there was a significant difference (Figure 5A).

For the second set of packages, the green package with a plant-based label (E7) demonstrated greater interest, with an average eye fixation of 3.58 s compared to the green package with a silicone label (E6), as presented in Figure 5. This observation indicates a significantly higher visual attraction for packaging with characteristics associated with sustainability. According to the results presented, there was a significant difference for the entire package, with a 5% significance level (Figure 5B).

For the third set of packages, the average attention retention was relatively close, with values of 3.15 for the white package with a plant-based label (E4) and 3.49 for the green package with a silicone label (E6). With a 5% significance level, the statistical analysis indicated that there was no significant difference between the views of the two packages (Figure 5C).

In the analysis of the last pair of packages, it was observed that the white package with a silicone label (E5) retained more attention, with an average time of 3.85, compared to the green package with a plant-based label (E7), which had an average time of 2.70. With a 5% significance level, this indicated a significant difference in attention retention between the two packages (Figure 5D).

In the analysis focused exclusively on the labels, we can observe the results with the average viewing time. The heatmap revealed where participants focused their gaze the most, with the red spot indicating greater gaze retention, highlighting the most impactful elements and the participant’s interest (Figure 6).

According to Figure 6, in the first image, E4 and E5, the term “plant-based” on the white packaging (E4) recorded higher visibility, with an average viewing time of 3.37 s. This indicates that consumers not only noticed the E4 packaging but also spent more time reading and recognizing the “plant-based” label. Through the heatmap, we can confirm which central elements of the E4 packaging were the most impactful and of greatest interest to the participants, as they are highlighted in red. Using the two-tailed test with a 5% significance level, a significant difference was observed in the attention retention of the labels. This result reinforces the conclusion that certain visual elements on the label have a greater ability to attract and retain consumer attention (Figure 6A).

In the second image of the analysis of attention retention on the labels of packages E6 and E7, the term “plant-based” on E7 achieved an average eye fixation time of 3.12 s, while the “silicone” label on E6 registered a lower average time of 2.68 s. In the heatmap, it is possible to identify that both images had similar eye fixation patterns, but the E7 package shows more intense central points. Through the two-tailed *t*-test with a 5% significance level, it was determined that there was no significant difference between the label views, suggesting that, statistically, the labels attracted a similar level of attention from the participants (Figure 6B).

In the third image of the analysis of attention retention on the labels of packages E4 and E6, it was observed that the term “silicone” on package E6 had a higher fixation time, with an average of 3.29 s, while package E4 with the “plant-based” label had an average of 2.89 s. Through the heatmap, we can observe greater visual retention intensity on package E4, despite the average fixation time being higher for the term “silicone” on package E6. Using the two-tailed test, we can confirm, with a 5% significance level, that there was no significant difference (Figure 6C).

In the analysis of Image 4, it was observed that the label with the term “plant-based” attracted more attention, achieving an average eye fixation time of 3.39 s, while the “silicone” label registered a lower average, with 2.34 s of eye fixation. Through the heatmap, a greater intensity of red can be seen on the E7 label with the term “plant-based”, corroborating the retention average result for the label. This can be confirmed through the statistical analysis of the two-tailed test with a 5% significance level, which indicated a significant difference in attention retention between the “plant-based” and “silicone” labels (Figure 6D).

Involving all 105 participants, a comparison was made between the white and green packaging and between the plant-based and silicone labels (Figure 7).

The results indicate an average attention retention of 3.38 s for the white packaging and 3.18 s for the green packaging, as presented in Figure 7A. This result suggests a slight preference or greater attention retention for white packaging compared to green packaging. Using a two-tailed statistical test with a 5% significance level, a significant difference was observed in attention retention between the white versus green packaging and labels with the terms plant-based versus silicone.

In the comparison between plant-based and silicone labels among all 105 participants (Figure 7B), the plant-based label achieved an average attention retention of 3.19 s, while the silicone label registered 2.67 s. Using a two-tailed statistical test with a 5% significance level, a significant difference was observed in attention retention between the white versus green packaging and labels with the terms plant-based versus silicone.

#### 3.5.1. Consumer Profile and Product Preference

The analysis was conducted with 105 participants (68.9% female, 31.1% male) with an average age of 23.5 years (Figure 8).

This study investigated consumer preferences for plant-based versus synthetic cosmetic products. Among the participants, 58.5% preferred plant-based products, 7.5% opted for synthetic products, and 34.0% had no specific preference, as shown in Figure 8.

These results underscore the growing demand for natural products and the importance of understanding consumer perceptions in cosmetic formulation.

Figure 9 compares consumer perceptions of synthetic and plant-based products regarding safety and efficacy, revealing several notable trends.

This study analyzed the reasons why consumers prefer plant-based cosmetics, based on 63 responses (Figure 9). Safety was the primary reason for 52.4% of participants, who believe that plant-based products contain fewer synthetic chemicals, which are seen as irritants. Additionally, 34.9% of participants stated that plant-based products are both safer and more effective, while only one participant cited efficacy as the main reason, emphasizing the importance of safety. Moreover, 11.1% selected “none of the alternatives”, suggesting personal preferences or environmental concerns.

This study evaluated consumers’ willingness to pay 20% more for cosmetics made from plant-based or synthetic ingredients. The results showed that 61% of participants prefer to pay more for plant-based products, mainly due to the perception of higher safety. In contrast, 39% are willing to pay more for synthetic products, which is possibly influenced by the perceived efficacy of synthetic products.

This study examined consumers’ perceptions of the environmental impact of raw materials used in cosmetics. Results indicate that 85.5% believe plant-based ingredients are less harmful to the environment than synthetic ones. Half of the participants consider synthetic materials more detrimental, while 3.2% acknowledge that plant-based materials can also be harmful. A minority see no direct link between the origin of raw materials and environmental impact, and a small fraction is uncertain about the issue.

The analysis of the final question regarding participants’ knowledge of sustainability shows that 89.6% believe that sustainability requires a balance between economic, social, and environmental aspects, and 71.7% acknowledge that synthetic products can also be sustainable.

This study explores consumer perceptions of sustainability in the cosmetic industry, highlighting the complexity of defining truly sustainable products. While most consumers (89.6%) recognize the need to balance economic, social, and environmental aspects, only 4.7% believe that all plant-based products are sustainable. The perception that biodegradability equals sustainability is held by 31.1% of participants. Furthermore, 22.6% view plant-based products as more sustainable. However, the majority (71.7%) acknowledge that synthetic products can also be sustainable.

#### 3.5.2. Packaging Preference Questionnaire Results

The results obtained from the eye-tracking test reveal an interesting variation in packaging preferences. The white packaging with a silicone label registered the highest preference, with a total of 60 preferences. This was closely followed by the green packaging with a plant-based label, which obtained 49 preferences. The green packaging, both with silicone and plant-based labels, showed similar preferences, with 46 and 49 respectively. The white packaging with a plant-based label presented the lowest preference, with only 14.

For this group of people, the perception of lower environmental impact is essential. Additionally, they prefer packaging where the color and label are aligned. For instance, if the packaging is white, they prefer labels that mention “silicone”. If the packaging is green, they prefer labels that mention “plant-based”. This demonstrates that the congruence between packaging color and label text influences their purchasing decisions, reflecting a preference for products that appear more natural and eco-friendlier.

## 4. Discussion

### 4.1. Rheological Analysis

According to Bonilha et al., 2020, rheological analysis plays an essential role in understanding the behavior of cosmetic formulations, directly influencing their application, stability, and user experience. Key properties such as thixotropy and pseudoplasticity are crucial for the development of effective formulations [10].

The analysis of rheological curves showed that most of the formulated emulsions exhibited thixotropic behavior, a property of certain fluids where their viscosity changes when subjected to mechanical stress or agitation, and then gradually returns to their original viscosity once the stress is removed. This behavior suggests that structural breakdown likely occurs during shear and can be related to the formulation [11]. The results show that Formulations B (polycitronellol HA), D (polycitronellol H), and G (solid petroleum jelly) showed the highest values of thixotropy, indicating the formation of more structured systems with a stronger network [12]. However, when the network breaks down, the viscosity of these formulations reduces, and they showed the lowest values of k when compared to the other formulations (Table 2). These rheological results can be related to the tendency of higher values of spreadability observed for these formulations in the sensorial analysis (Table 3, Table 4 and Table 5).

Formulations A (base), E (Dimethicone 350), and F (polycitronellol compound) exhibited lower values of thixotropy and similar n values (ranging between 0.44 and 0.46), indicating that they have similar pseudoplastic behavior. In other words, these formulations do not exhibit a linear viscosity response to shear rate, as observed in Newtonian fluids (which maintain constant viscosity regardless of shear rate). Instead, their viscosity decreases sharply as the shear rate increases. This occurs due to the orientation of particles or molecules within the fluid, which align in the direction of flow under force, reducing internal resistance and, consequently, viscosity [13], which facilitates product application, distribution on the skin, and flow from the packaging [14]. Nevertheless, these formulations showed the highest values of k and η_100s_^−1^, which can be related to the tendency of lower values of spreadability (Table 3, Table 4 and Table 5).

Formulation C (dimethicone 100) was the only formulation that did not exhibit thixotropy and it showed higher values of n than most formulations, indicating that this viscosity is less affected by the shear rate.

Studies such as those by Jeong et al., 2017, emphasize the importance of these rheological properties in developing cosmetic products that are not only stable and effective but also sensorially appealing. These characteristics ensure that formulations meet desired functionality, ease of application, and user satisfaction [15].

### 4.2. Sensory Analysis

The analysis of sensory attributes evaluated in the cosmetic formulations highlights significant differences that directly influence consumer perception and acceptance of these products. Attributes such as dry touch, velvety film, residual oiliness, spreadability, and absorption are critical for the commercial and functional success of skincare products. Previous studies emphasize that the combination of functionality and a pleasant sensory experience is essential to meet the expectations of the target audience and promote product loyalty [16]

The results indicate that Formulation B, containing polycitronellol (HA), exhibited higher levels of dry touch and velvety film, characteristics strongly associated with smoothness and comfort during application. In contrast, Formulation C, with dimethicone 100, demonstrated higher residual oiliness. This behavior aligns with previous studies showing that long-chain silicones, such as dimethicone, tend to form a thicker film on the skin, enhancing the perception of oiliness; e.g., see Decker et al., 2023 [17]. However, the performance of Formulation B (polycitronellol HA) in these sensory attributes should also be evaluated through the lens of sustainability. While silicones such as dimethicone are widely used for their sensory efficacy and stability, their environmental persistence is a major concern. Organic compounds derived from renewable sources, such as polycitronellol HA, have proven to be promising alternatives due to their biodegradability and lower environmental impact, aligning with the increasing demand for eco-friendly products [18].

Another important sensory attribute evaluated was adhesiveness, which was notably higher in Formulation D, containing polycitronellol H. Excessive adhesiveness, often associated with a “sticky” sensation, can negatively affect product acceptance, particularly for daily-use applications. Formulations with balanced adhesiveness are generally better accepted by consumers [19]. To mitigate this effect, adjustments in the emollient concentration or combinations with other ingredients, such as texturizing agents or natural polymers, can be effective and environmentally conscious strategies. Formulations with synthetic ingredients, such as dimethicone, although demonstrating lower adhesiveness, have a higher environmental impact due to challenges in biodegradability and high energy demands during production. Conversely, polycitronellol offers a more sustainable alternative, delivering desirable sensory attributes such as dry touch and velvety finish while minimizing environmental impact [20].

Spreadability, on the other hand, was an attribute that showed consistency across formulations, suggesting that the choice of emollients effectively ensured uniform and pleasant application. This uniformity can facilitate the substitution of less sustainable ingredients with ecological alternatives without compromising sensory performance.

Another relevant sensory aspect is absorption time. The literature indicates that the perception of a quick “absorption point” is crucial to increasing the acceptance of daily-use products, as consumers prefer formulations that penetrate the skin quickly without leaving noticeable residues [21,22]. However, Formulation D (polycitronellol H) demonstrated higher absorption levels compared to Formulation E (dimethicone 350).

The impact of shine and residual oiliness was more noticeable in Formulations F (polycitronellol compound) and G (solid petrolatum), which were richer in oily components. These attributes can be advantageous for products with barrier or hydration functions but present limitations for products intended for oily skin or use in hot climates [23]. On the other hand, the uniformity in attributes such as spreadability, glide, and white residue among formulations suggests that these properties were consistent and adequate regardless of composition.

The selection of emollients, such as polycitronellol and dimethicones, must consider not only their sensory properties but also their impact on sustainability perception. Polycitronellol shows greater compatibility with sustainable practices, positioning itself as an innovative solution to meet the demands of increasingly environmentally conscious consumers. Furthermore, the selection of sustainable ingredients should not be limited to their direct environmental impact but also consider ethical and economic aspects, promoting circular economies and regional development [24].

### 4.3. Evaluation of Hydration and Transepidermal Water Loss (TEWL)

These results suggest that certain formulations provide distinct hydration levels over time, which can be attributed to their specific characteristics, such as composition and the type of ingredients used. Identifying these differences is crucial to understanding how various formulations can meet the skin’s hydration needs, thereby influencing consumer product choices.

The methodology used and the results obtained reinforce the importance of rigorous testing to evaluate the efficacy of cosmetic products, ensuring that formulations effectively meet users’ hydration expectations. These insights can guide future product development and improve recommendations for consumers based on scientific evidence.

Regarding the products, Formulations B (polycitronellol HA) and C (dimethicone 100) demonstrated that, despite the decrease in hydration observed in Formulation A (base), the presence of emollients was able to maintain the skin’s initial hydration, suggesting effective action for skin hydration. Furthermore, it was found that Formulation B (polycitronellol HA) showed a significant improvement in hydration during the first two hours compared to Formulation A (base), suggesting that substitution with the sustainable polymer may offer advantages over the tested silicone.

According to the study by McIntosh et al., 2018., which evaluated the performance of algae-derived alkenones as a green alternative to waxes in cosmetics, the potential role of sustainable and naturally derived ingredients in the cosmetics industry is emphasized. Such ingredients can not only contribute to skin hydration but also align cosmetic products with consumer expectations regarding sustainability [25].

These results have significant implications for the development of cosmetic products focused on skin hydration and barrier protection. The choice of ingredients should consider not only their occlusive properties but also their ability to enhance skin hydration and provide a pleasant sensation after application.

Furthermore, cosmetic formulations should balance efficacy and sensoriality, ensuring that products not only offer clinical benefits but also meet consumer preferences in terms of texture and user experience.

The importance of texture as a decisive factor in the perception and acceptance of cosmetic products is frequently highlighted in the literature, as noted by [26].

Pham et al., 2021, show that hydration is crucial as it increases the water content in the stratum corneum, enabling the mobility of lipid and protein components, which is essential for the integrity and function of the skin barrier [27].

Cosmetic formulations should balance efficacy and sensoriality, ensuring that products not only provide clinical benefits but also meet consumer preferences in terms of texture and user experience.

Kottner et al., 2013, demonstrated in their studies that TEWL varies significantly across different body sites, such as the palms, soles, armpits, forehead, and forearm, showing higher values. This increase in TEWL in certain areas is associated with the low sebaceous lipid content in these regions [28].

According to Scheel-Sailer et al., 2015, abnormal skin hydration, whether elevated or decreased, can reduce skin resistance and increase the risk of infection. Therefore, maintaining adequate hydration is essential for the health and integrity of the skin barrier [29].

These results are significant for the development of cosmetic products, emphasizing the importance of formulations that not only hydrate the skin but also strengthen its natural barrier. The efficacy of formulations in maintaining low TEWL levels is crucial for products intended for sensitive skin or conditions that require enhanced skin barrier protection.

The choice of ingredients and the formulation of cosmetic products should, therefore, be carefully considered to ensure they provide both hydrating and barrier-protective benefits, thus meeting consumer needs and contributing to the health and integrity of the skin.

According to Plopski et al., 2022, a lower TEWL factor indicates higher water content and a more intact skin barrier. Formulations B and C demonstrated lower TEWL values than Formulation A, suggesting that both polycitronellol HA polymer and silicone oil provide effective protection against water loss [30].

In the hydration analysis, Formulation B achieved better average results compared to Formulation C. However, in the TEWL analysis, no significant differences were observed between Formulations B and C, with both showing similar average values.

These results are crucial for the development of cosmetic products focused on skin hydration and barrier protection. The choice of ingredients should consider their occlusive properties and ability to improve skin hydration, while also providing a pleasant sensory experience after application.

### 4.4. Neurosensory

Eye-tracking analysis provides valuable insights into how the visual characteristics of packaging influence consumer perception and purchasing decisions. This result suggests that packaging with a plant-based appeal tends to hold consumers’ attention for longer, which may indicate a preference or greater interest in products perceived as more natural or sustainable.

Eye-tracking technology allows researchers to precisely measure where and for how long consumers focus on different areas of a package, providing crucial insights into the dynamics of attention and consumer behavior [31]. The color of the packaging plays a significant role in capturing consumer attention, with white packaging showing slightly higher retention rates [32]. Although the differences in attention retention between the two colors are not drastic, they are significant, suggesting that even subtle changes in design can significantly impact consumer engagement [32].

The higher standard deviation observed in the green packaging indicates a more varied response among consumers, which may be attributed to individual preferences or the influence of color psychology [33].

This tool is particularly useful for identifying which aspects of the label most effectively capture consumer attention, allowing designers and marketing professionals to adjust visual elements to maximize the visual impact and appeal of products on shelves.

The use of heat maps as a complement to the analysis of fixation time provides a more detailed understanding of consumer behavior and how graphic elements can be optimized to enhance visual communication and packaging design effectiveness.

This information is essential in understanding that, despite possible visual variations among the labels, none stood out significantly in terms of attracting more consumer attention within the established margin of significance. This finding can guide future label design strategies, highlighting the need to explore other elements or techniques that can effectively differentiate a product visually in competitive environments.

The detailed analysis of label visualizations is crucial for developing packaging that not only captures attention but also sustains consumer interest, thus enhancing brand perception and influencing purchasing decisions.

Studies such as those by Proi et al., 2023, have demonstrated that eco-labels have a significant influence on consumer attention and decision-making at the point of sale, highlighting the importance of sustainable product characteristics [34]. Furthermore, the adoption of sustainable practices and labeling in marketing is not only a response to consumer demand but also a driving force behind the growing interest in eco-friendly products [35].

The role of visual elements in eco-labels, such as size and relevance, is crucial, as they significantly affect consumer attention and product choice [36]. The greater variation in attention retention for “plant-based” products suggests a wider range of consumer engagement, which may be influenced by individual perceptions of sustainability and product content [37].

These results suggest that characteristics associated with sustainability and the natural origin of products, as indicated by the term “plant-based”, have a significant impact on attracting consumer attention. This trend reflects a growing consumer preference for products perceived as more ecological or beneficial for health.

#### 4.4.1. Consumer Profile and Product Preference

A study by Meier et al., 2019, explored consumer preferences for natural versus synthetic products in the pharmaceutical industry, aligning with findings in the cosmetics sector. Consumers demonstrated a strong preference for natural products, even when safety and efficacy were identical. This preference is primarily driven by perceptions of greater safety and reduced use of synthetic chemicals in natural products, which are often seen as potential irritants or allergens [38]. Similarly, a study in Poland found that 32% of participants identified fewer chemical components as a key factor making natural cosmetics safer [39]. Plant-based products are increasingly viewed as both safe and effective. Alves et al., 2020, demonstrated the antioxidant and anti-inflammatory properties of plant extracts, which benefit skin health [40]. Additionally, sustainability concerns, such as the environmental impact of cultivating and processing plant-based ingredients, have been highlights in research, reinforcing their importance in the cosmetics industry [20]. Regarding the willingness to pay more for plant-based products, 61.3% of respondents in the study by Ratajczak et al., 2023, indicated they would pay a premium, aligning with findings from this study [38]. Ethical and environmental values significantly influenced these choices, emphasizing the perceived environmental benefits of plant-based materials. However, divergent opinions and uncertainties suggest a need for better consumer education on the environmental impact of raw materials. Survey results indicate a strong preference for plant-based over synthetic raw materials due to their perceived lower environmental impact. Teixeira et al., 2022, highlight the trend of upcycling in the cosmetics industry, demonstrating increased consumer awareness of product origin and environmental impact. While synthetic products are often seen as more effective, plant-based options are favored for their safety and sustainability, reflecting a balance influenced by efficacy, safety, and ethical considerations. The preference for sustainable products indicates a shift in consumer behavior, with buyers considering environmental impacts alongside efficacy and safety [41]. This underscores the need for transparent production practices and the effective communication of sustainability efforts to meet consumer expectations. Recent studies, such as Rocca et al., 2022, emphasize the demand for eco-friendly alternatives to traditional ingredients such as synthetic surfactants and silicone oils [42]. Lima et al., 2023, also stress the importance of botanical formulations with lower environmental and health impacts [22]. Despite the benefits of biodegradable packaging, Siddiqui et al., 2023 [43], caution that many such materials still rely on unsustainable additives, highlighting the complexity of defining true sustainability. Material innovations, such as starch-based nanocomposites, offer promising eco-friendly solutions for packaging in the cosmetics sector [18]. Interestingly, 22.6% of participants viewed plant-based products as more sustainable, while 71.7% acknowledged the sustainability potential of synthetic products. This aligns with the findings of Moshood et al., 2021, which emphasize the growing market and policy shifts favoring biodegradable plastics. The findings reflect the complexity of consumer attitudes toward sustainability, emphasizing the need for clear communication and education about sustainable practices. An integrated approach to production and policy is essential to align cosmetic products with genuine sustainability principles. Addressing these concerns is critical for meeting the expectations of increasingly conscious consumers while promoting environmentally responsible practices in the cosmetics industry [44].

#### 4.4.2. Packaging Preference Questionnaire Results

These results may reflect not only consumers’ perceptions of sustainable content, as indicated by vegetable labels, but also other factors such as color associations, which have psychological and cultural implications influencing purchasing decisions.

In the context of cosmetics, consumer perception and satisfaction can be influenced by a wide range of factors, from perceived efficacy to the sensory appeal of the product. A study by Susanti Kunda and Heti Tri Rahmawati, 2022, explored the perception of price, product quality, brand ambassadors, and consumer loyalty, with consumer satisfaction mediating these [45].

These findings are applicable for understanding how different attributes of cosmetic products influence average evaluations and variation in these evaluations among consumers. Consumer behavior studies using eye tracking offer valuable insights into how people perceive and process information related to sustainability in product packaging. For example, Chiu et al., 2023, investigated how the visual perception of reusable packaging and logo type simplification affect consumer sustainable awareness. They found that reusable packaging can enhance both attention and attraction, indicating that sustainable packaging tends to draw more attention and, consequently, can influence purchase decisions [46].

Mehta et al., 2023, examined how the physical characteristics of packaging affect consumer attention during initial product interaction. The study revealed that packaging that attracts more attention does not always score highest in likability, demonstrating a disconnect between what attracts visual attention and what is emotionally preferred or considered important [47].

This suggests that in the context of synthetic and vegetable cosmetic packaging, perceived sustainability through design and presented information can significantly impact consumer purchasing behavior. Vegetable-labeled packaging, which scored higher in the mentioned study, may be perceived as more aligned with sustainability values, while silicone-labeled packaging, which scored lower, may not evoke the same perception of environmental responsibility.

These findings are important for companies and packaging designers seeking to promote sustainable products. Packaging design should be carefully considered to ensure that sustainability credentials are clearly communicated and easily perceptible using eye tracking and other behavioral research methods. Thus, it is possible to develop packaging that not only attracts consumer attention but also effectively communicates the sustainability message and positively influences purchasing behavior.

Eye-tracking technology is increasingly being investigated for its potential in various screening processes due to its precise measurement capabilities, further emphasizing its utility in marketing research; e.g., see Kanhirakadavath et al., 2022 [48].

## 5. Conclusions

The results of this study indicate that emulsions formulated with sustainable homopolymers demonstrate satisfactory moisturizing properties, sensory characteristics approved by evaluators, and good acceptance by the public. Furthermore, all the tests conducted suggest that the replacement of traditional emollients with sustainable alternatives can be made without affecting consumer experience. This means that the same benefits can be delivered, with the additional advantage of being sustainable. This approach not only meets consumer expectations but also aligns with industry’s commitment to environmental responsibility and innovation.

## Figures and Tables

**Figure 1 pharmaceutics-17-00265-f001:**
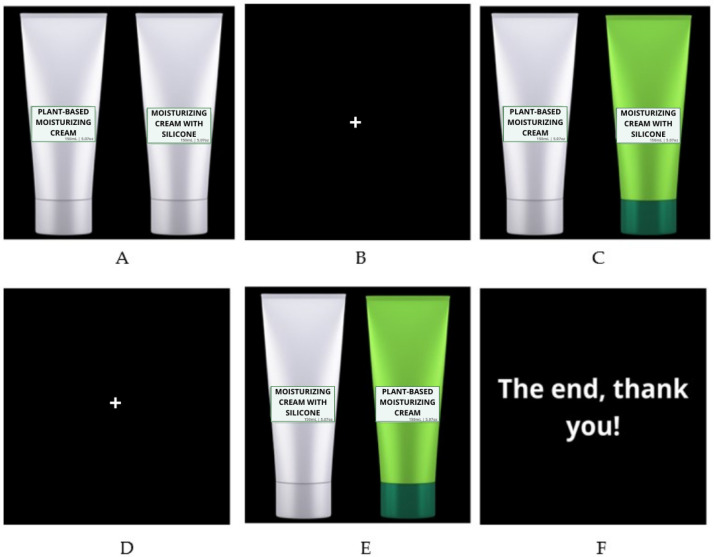
Timeline presented in eye tracking. (**A**) White packaging with the label “Plant-based Moisturizing Cream”—E4 and white packaging with the label “Moisturizing Cream with Silicone”—E5. (**B**) Image for eye rest. (**C**) White packaging with the label “Plant-based Moisturizing Cream”—E4, and green packaging with the label “Moisturizing Cream with Silicone”—E6. (**D**) Image for eye rest. (**E**) White packaging with the label “Moisturizing Cream with Silicone”—E5, and green packaging with the label “Plant-based Moisturizing Cream”—E7. (**F**) Thank you screen and conclusion of the eye-tracking analysis.

**Figure 2 pharmaceutics-17-00265-f002:**
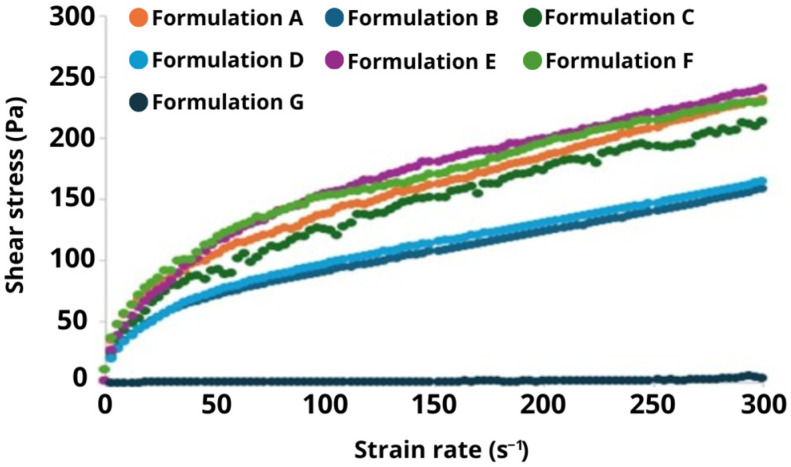
Flow curves of the formulations in steady state.

**Figure 3 pharmaceutics-17-00265-f003:**
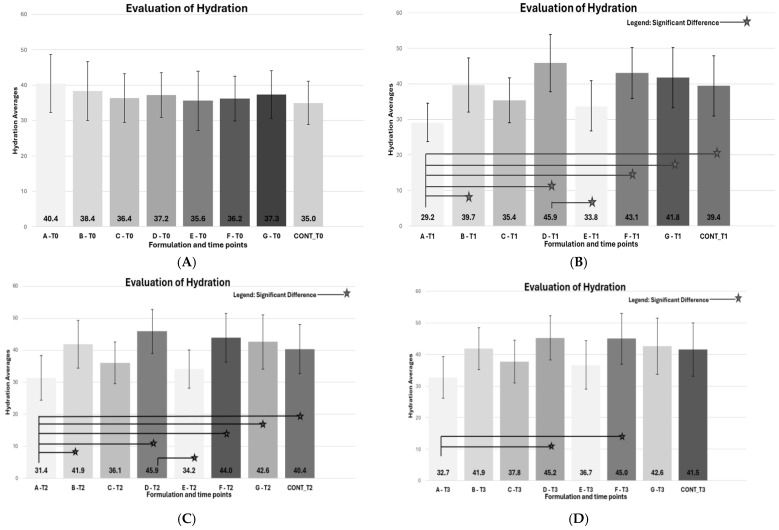
In vivo hydration test at times T0, T1, T2, and T3. (**A**) Average results of the in vivo hydration test at time T0. (**B**) Average results of the in vivo hydration test at time T1 with significant differences demonstrated by the symbol presented in the legend. (**C**) Average results of the in vivo hydration test at time T2 with significant differences demonstrated by the symbol presented in the legend. (**D**) Average results of the in vivo hydration test at time T3 with significant differences demonstrated by the symbol in the figure.

**Figure 4 pharmaceutics-17-00265-f004:**
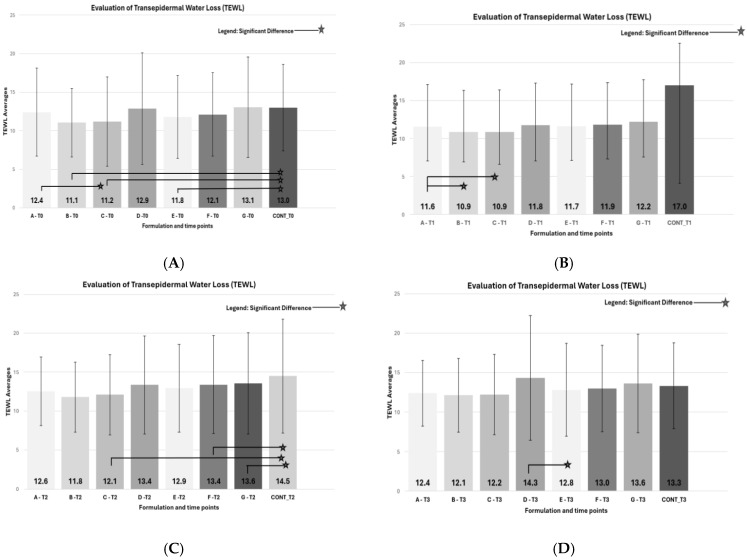
In vivo transepidermal water loss TEWL test at times T0, T1, T2, and T3. (**A**) Average results of the in vivo TEWL test at time T0, with significant differences indicated by symbols according to the legend. (**B**) Average results of the in vivo TEWL test at time T1, with significant differences presented in the figure. (**C**) Average results of the in vivo TEWL test at time T2, with significant differences demonstrated in the figure. (**D**) Average results of the in vivo TEWL test at time T3, with significant differences demonstrated in the figure.

**Figure 5 pharmaceutics-17-00265-f005:**
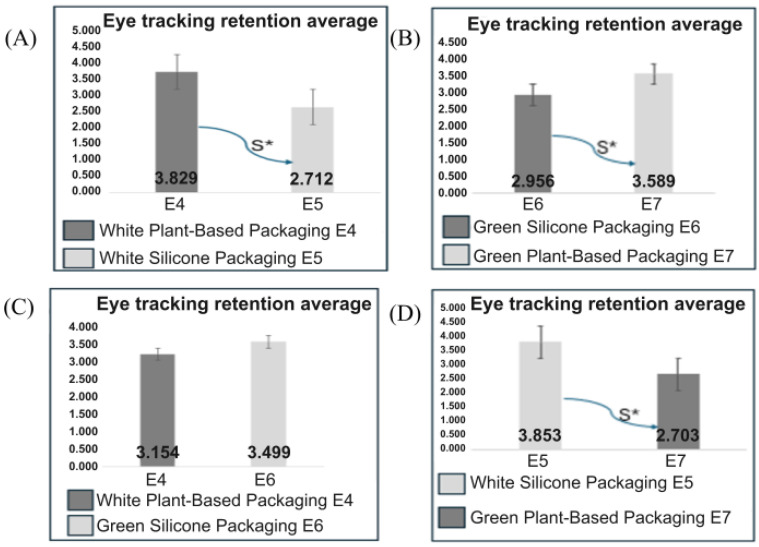
Eye tracking retention average E4 vs. E5, E6 vs. E7, and E4 vs. E6. (**A**) Average attention retention from eye tracking analyzing packages E4 and E5, showing a significant difference as indicated by the symbol according to the legend. (**B**) Average attention retention from eye tracking analyzing packages E6 and E7, showing a significant difference as indicated by the symbol according to the legend. (**C**) Average attention retention from eye tracking analyzing packages E4 and E6, which did not show a significant difference. (**D**) Average attention retention from eye tracking analyzing packages E5 and E7, showing a significant difference as indicated by the symbol according to the legend.

**Figure 6 pharmaceutics-17-00265-f006:**
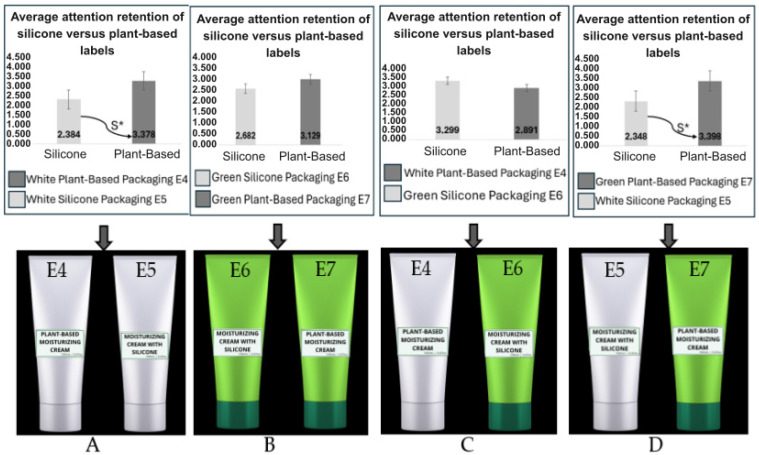
Average attention retention of silicone vs. plant-based label heatmap of packages. (**A**) Average attention retention of the silicone vs. plant-based label and heatmap of visual retention for the E4 and E5 packages, showing a significant difference, as indicated by the symbol according to the legend. (**B**) Average attention retention of the silicone vs. plant-based label and heatmap of visual retention for the E6 and E7 packages. (**C**) Average attention retention of the silicone vs. plant-based label and heatmap of visual retention for the E4 and E6 packages. (**D**) Average attention retention of the silicone vs. plant-based label and heatmap of visual retention for the E5 and E7 packages, showing a significant difference, as indicated by the symbol according to the significant difference.

**Figure 7 pharmaceutics-17-00265-f007:**
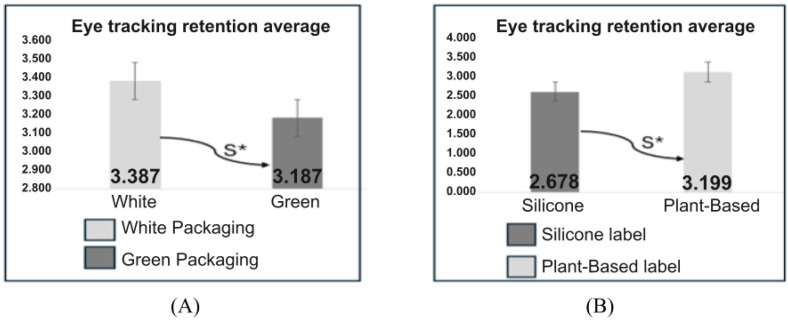
Averages of attention retention of white versus green packaging and for the plant-based and silicone labels among 105 participants. (**A**) Average attention retention of white versus green packaging showed a significant difference, as indicated by the symbol according to the legend. (**B**) Average attention retention of plant-based versus silicone labels, showed a significant difference, as indicated by the symbol according to the legend.

**Figure 8 pharmaceutics-17-00265-f008:**
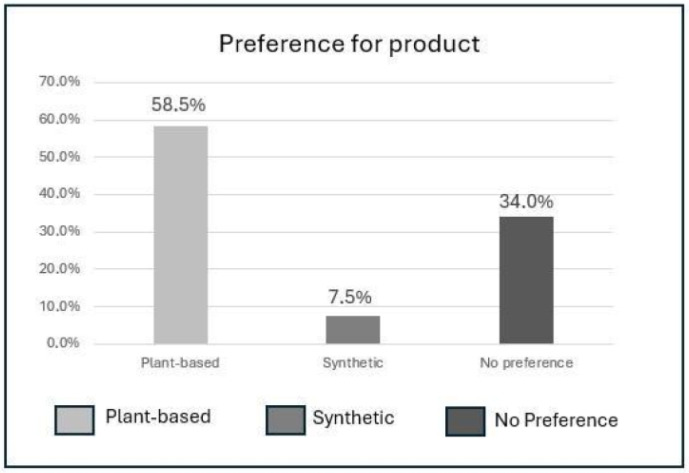
Preference for plant-based vs. synthetic products.

**Figure 9 pharmaceutics-17-00265-f009:**
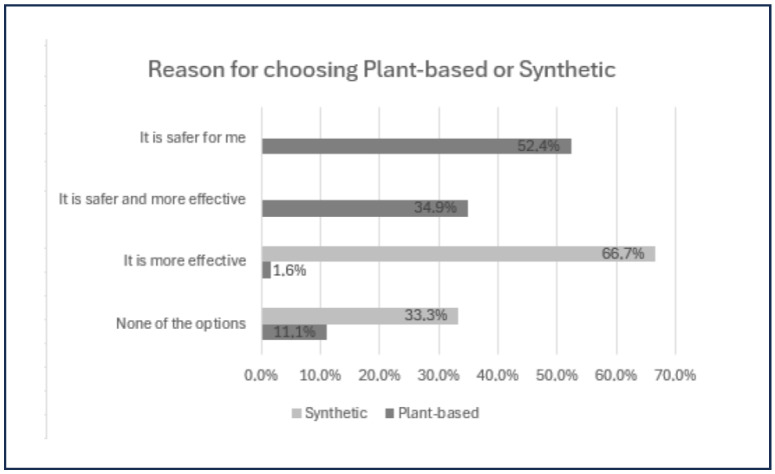
Reason for choosing plant-based vs. synthetic products.

**Table 1 pharmaceutics-17-00265-t001:** Composition of seven emulsions developed for this study, including INCI names and suppliers of ingredients.

INCI/Raw Material	Supplier	FormulationA	FormulationB	Formulation C	FormulationD	FormulationE	FormulationF	FormulationG
Cetearyl Alcohol	AQIA	6 g	6 g	6 g	6 g	6 g	6 g	6 g
Glyceryl Stearate	AQIA	2 g	2 g	2 g	2 g	2 g	2 g	2 g
Ceteareth-20	AQIA	3.5 g	3.5 g	3.5 g	3.5 g	3.5 g	3.5 g	3.5 g
Sorbitan Stearate	AQIA	1.4 g	1.4 g	1.4 g	1.4 g	1.4 g	1.4 g	1.4 g
Polycitronellol Acetate (HA)	P2Science	—	10 g	—	—	—	—	—
Dimethicone (100)	Labsynth	—	—	10 g	—	—	—	—
Polycitronellol (H)	P2Science	—	—	—	10 g	—	—	—
Dimethicone (350)	Labsynth	—	—	—	—	10 g	—	—
Polycitronellol andEuphorbia Cerifera Candelilla Wax	P2Science	—	—	—	—	—	10 g	—
Petrolatum	Labsynth	—	—	—	—	—	—	10 g
Phenoxyethanol	Proserv	0.4 g	0.4 g	0.4 g	0.4 g	0.4 g	0.4 g	0.4 g
Water	—	86.7 g	76.7 g	76.7 g	76.7 g	76.7 g	76.7 g	76.7 g

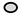
 Polycitronellol Acetate HA (Formulation B) as a potential substitute for dimethicone 100 (Formulation C). 
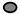
 Polycitronellol H (Formulation D) as a potential substitute for dimethicone 350 (Formulation E). 
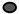
 Polycitronellol and Euphorbia cerifera (candelilla wax) (Formulation F) as a potential substitute for solid petrolatum (Formulation G).

**Table 2 pharmaceutics-17-00265-t002:** Rheological parameters obtained from the flow curves.

Formulation	Thixotropy (Pa/s)	σ_0_ (Pa)	k (Pa.sn)	N	η_1005_^−1^ (mPa.s)
A (Base)	5234.7 ± 653.8	0	17.07 ± 3.59	0.45 ± 0.02	1329.8 ± 133.6
B (Polycitronellol HA)	14,122.2 ± 104.6	0	12.31 ± 0.14	0.44 ± 0.00	910.7 ± 7.1
C (Dimethicone 100)	0	0	11.50 ± 4.72	0.51 ± 0.06	1101.9 ± 230.7
D (Polycitronellol H)	10,559.0 ± 922.3	0	15.01 ± 3.05	0.44 ± 0.01	1099.2 ± 177.3
E (Dimethicone 350)	6009.9 ± 329.3	9.91 ± 5.63	18.32 ± 3.22	0.45 ± 0.02	1477.0 ± 115.5
F (Polycitronellol Compound)	8173.8 ± 411.9	0	17.42 ± 4.06	0.46 ± 0.05	1539.6 ± 65.4
G (Solid Petrolatum)	10,443.0 ± 1634.5	0	0.04 ± 0.01	0.74 ± 0.08	10.22 ± 0.70

Legend: Emulsion formulations A, B, C, D, E, F, G.

**Table 3 pharmaceutics-17-00265-t003:** Means, standard deviations (S.D.), and Fisher’s LSD test results for samples A-B-C. Means followed by the same letter (A or B) in each row do not differ significantly from each other at a 5% significance level.

Attribute	Formulation A (Base)		Formulation B (Polycitronellol HA)		Formulation C (Dimethicone 100)				
	Mean Intensity	Standard Deviation	Mean Intensity	Standard Deviation	Mean Intensity	Standard Deviation	F-Test Value	*p*-Value	Levene’s Test for Homogeneity of
Spreadability	5.69 A	0.26	5.71 A	0.20	5.62 A	0.22	7.01	0.004	0.272
Glide	5.70 A	0.30	5.73 A	0.21	5.65 A	0.24	5.52	0.010	0.514
Stickiness	0.63 A	0.36	0.55 A	0.33	0.71 AB	0.40	1.60	0.222	0.756
Residue	0.01 A	0.01	0.01 A	0.01	0.01 A	0.02	0.26	0.775	0.599
Residual White Residue	0.01 A	0.01	0.00 A	0.00	0.01 A	0.02	1.96	0.161	0.634
Velvety Film	2.36 A	0.36	2.18 A	0.46	2.01 B	0.36	0.84	0.442	0.702
Dry Touch	8.64 A	0.42	8.69 A	0.33	8.14 B	0.71	8.57	0.001	0.149
Immediate Shine	4.70 A	0.48	4.78 A	0.54	4.81 A	0.53	0.18	0.839	0.785
Residual Shine	2.20 B	0.36	2.35 B	0.43	2.23 B	0.60	16.74	1 × 10^−4^	0.107
Immediate Oiliness	1.20 A	0.42	1.19 A	0.33	1.28 A	0.52	0.94	0.403	0.807
Residual Oiliness	0.17 A	0.16	0.39 A	0.60	0.48 A	0.60	1.46	0.252	0.442
Immediate Greasy Film	1.78 A	0.33	1.77 A	0.31	1.67 A	0.53	0.15	0.862	0.299
Residual Greasy Film	0.29 B	0.18	0.34 B	0.22	0.47 A	0.36	6.26	0.006	0.328
Absorption point	8.33 A	1.03	8.11 A	0.90	8.00 A	1.12	1.32	0.284	0.810

**Table 4 pharmaceutics-17-00265-t004:** Means, standard deviations (S.D.), and Fisher’s LSD test results for samples A-D-E. Means followed by the same letter (A or B) in each row do not differ significantly from each other at a 5% significance level.

Attribute	Formulation A (Base)		Formulation D (Polycitronellol H)		Formulation E (Dimethicone 350)				
	Mean Intensity	Standard Deviation	Mean Intensity	Standard Deviation	Mean Intensity	Standard Deviation	F-Test Value	*p*-Value	Levene’s Test for Homogeneity of
Spreadability	5.69 A	0.26	5.70 A	0.23	5.62 A	0.22	0.71	0.500	0.641
Glide	5.70 A	0.30	5.54 A	0.59	5.65 A	0.24	0.74	0.488	0.665
Stickiness	0.63 B	0.36	1.03 A	0.45	0.71 B	0.40	6.58	0.005	0.673
Residue	0.01 A	0.01	0.01 A	0.01	0.01 A	0.02	0.29	0.748	0.856
Residual White Residue	0.01 A	0.01	0.01 A	0.01	0.01 A	0.02	0.46	0.635	0.897
Velvety Film	2.36 A	0.36	1.86 B	0.32	2.01 B	0.36	8.38	0.002	0.396
Dry Touch	8.64 A	0.42	8.26 B	0.35	8.14 B	0.71	4.97	0.015	0.635
Immediate Shine	4.70 A	0.48	4.87 A	0.54	4.81 A	0.52	0.93	0.407	0.155
Residual Shine	2.20 B	0.36	2.72 A	0.46	2.23 B	0.60	9.19	0.001	0.029
Immediate Oiliness	1.20 A	0.42	1.13 A	0.40	1.28 A	0.52	1.27	0.298	0.603
Residual Oiliness	0.17 B	0.16	0.35 AB	0.29	0.48 A	0.60	2.48	0.103	0.310
Immediate Greasy Film	1.78 A	0.33	1.71 A	0.43	1.67 A	0.53	0.22	0.804	0.545
Residual Greasy Film	0.29 B	0.18	0.50 A	0.30	0.47 A	0.35	5.77	0.008	0.635
Absorption point	8.33 A	1.03	8.32 A	1.50	7.68 B	1.08	8.10	0.002	0.039

**Table 5 pharmaceutics-17-00265-t005:** Means, standard deviations (S.D.), and Fisher’s LSD test results for samples A-F-G. Means followed by the same letter (A or B) in each row do not differ significantly from each other at a 5% significance level.

Attribute	Formulation A (Base)		Formulation F (Polycitronellol Compound)		Formulation G (Solid Petrolatum)				
	Mean Intensity	Standard Deviation	Mean Intensity	Standard Deviation	Mean Intensity	Standard Deviation	F-Test Value	*p*-Value	Levene’s Test for Homogeneity of
Spreadability	5.69 A	0.26	5.52 A	0.20	5.70 A	0.23	1.37	0.272	0.299
Glide	5.70 A	0.30	5.69 A	0.21	5.72 A	0.23	0.08	0.920	0.118
Stickiness	0.63 B	0.36	1.04 A	0.33	0.76 B	0.44	5.36	0.011	0.890
Immediate White Residue	0.01 A	0.01	0.01 A	0.01	0.01 A	0.01	0.08	0.922	0.286
Residual White Residue	0.01 A	0.01	0.00 A	0.00	0.02 A	0.06	0.97	0.392	0.566
Velvety Film	2.36 A	0.36	2.16 AB	0.46	2.05 B	0.29	3.80	0.036	0.661
Dry Touch	8.64 A	0.42	8.23 B	0.33	8.21 B	0.46	5.80	0.008	0.604
Immediate Shine	4.70 A	0.48	4.84 A	0.54	4.93 A	0.69	0.71	0.500	0.576
Residual Shine	2.20 B	0.36	2.80 A	0.43	2.76 A	0.64	11.01	0.000	0.637
Immediate Oiliness	1.20 A	0.42	1.20 A	0.33	1.23 A	0.28	0.06	0.940	0.855
Residual Oiliness	0.17 A	0.16	0.57 A	0.60	0.38 A	0.56	1.56	0.230	0.243
Immediate Greasy Film	1.78 A	0.33	1.86 A	0.31	1.90 A	0.30	0.71	0.503	0.847
Residual Greasy Film	0.29 B	0.18	0.63 A	0.22	0.59 A	0.59	3.78	0.036	0.655
Absorption point	8.33 AB	1.03	8.00 B	1.31	8.55 A	1.39	3.49	0.045	0.781

## Data Availability

The original contributions presented in this study are included in the article. Further inquiries can be directed to the corresponding author.

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
