# Peer review of "Physicochemical and Sensory Evaluation of Sustainable Plant-Based Homopolymers as an Alternative to Traditional Emollients in Topical Emulsions"

_pharmaceutics, 2025, doi:10.3390/pharmaceutics17020265_

Round 1

Reviewer 1 Report

Comments and Suggestions for Authors

I attached the manuscript with my comments and suggestions.

Author Response

Dear Reviewer 1,

The responses to your valuable comments are included in the attached document.

Thank you very much.

Reviewer 2 Report

Comments and Suggestions for Authors

This review concerns the article type manuscript entitled “Physicochemical and Sensory Evaluation of Sustainable plant-based Homopolymers as an Alternative to Traditional Emollients in Topical Emulsions” and submitted to Pharmaceutics.
The work presents the study on seven (7) formulations, denoted as A-G. The materials used for the formulation in this work were: cetearyl alcohol, glyceryl stearate, ceteareth-20, sorbitan stearate, phenoxyethanol, which are common for all formulations, as well as polycitronellol acetate HA (B), dimethicone 100 (C), polycitronellol (D), dimethicone 350 (E),  Polycitronellol and Euphorbia Cerifera wax (F), Petrolatum 100 (G), and no extra material (A). The formulations were investigated using rheological analysis, sensory analysis with a trained panel, hydration and transepidermal water loss test, neurosensory analysis. Participants (105), both male and female, aged between 18 and 40 years were involved.
The title of the manuscript is consistent with the main text.
The aim of the work is consistent with the scope of the journal.
In my opinion, the manuscript can be considered for publication.
Nevertheless, the following remarks should be considered in the revised version.
Abstract
Objectives: It is not explained what kind of plant-based homopolymers are investigated.
Materials and Methods: Chapter 2.2. Emulsion mixing speed not given. What type of mixer was used?
Results: Show hard evidence what “excellent viscoelastic stability” as well as “hydration and moisture retention” means.
Table 1. The table is presented in not editing way (JPG or PNG, pixels).
Table 1. Decimal data should be after the dot.
Table 2. The wording marked with the letters A-G does not make it easier to read. The distinctive component should be mentioned.
Using the expression in the text, e.g. formulation G, does not make reading easier. I propose, formulation G (petrolatum).
Figure 4 is a table, not figure.
Figures 5-9. Description of the significant difference should be incorporated in the figure, not in legend.
Figure 10. Low quality
In the statistical analysis, please provide the F-test value for the Anova analysis. What test values were obtained to check for homogeneity of variance.
Figures 5 and 6. Why is a different scale used for A (Y-axis) than for B-D? Compare the time-dependent results (Figure 5) in one graph, i.e. T0-T3. Use the same approach for Figure 6.
In the TEWL and hydration analysis, the number of measurements (n variables) is not given.
All error bars in Figure 5 are the same. A similar situation occurs in other graphs. Please explain.
Minor remarks:
na evaluation
75°C (nits separately)
s^-1
physicochemical Properties
sensory Properties
[24] demonstrated
rates [28] Although
[36] demonstrated
[38] highlights
[46] examined
research[47]

Author Response

Dear Reviewer 2,

The responses to your valuable comments are included in the attached document.

Thank you very much.

Reviewer 3 Report

Comments and Suggestions for Authors

In this work, authors elucidated the advantages and preferences of formulation made through sustainable homopolymers over traditional emollients. The study involves a wide aspect, including formulation, rheological analyses, hydration measurement, sensory evaluations, and neurosensory analysis. However, the whole manuscript provides too many results that some of them are not helpful to strengthen the depth of this manuscript. Instead, the experimental designs and results discussions should be more detailed to bridge the formulation characterization results with the sensory evaluations. Therefore, this paper is suggested to have major revisions. Otherwise, I’m afraid it won't be suitable for  Pharmaceutics Journal. Detailed comments are listed as follows:

1.      On page 7, the characteristics of Formulation G being stress overshoot at low shear rate is not clearly presented from Figure 2.

2.      Sensory evaluation in Figure 4: Means followed by the same letter (A or B) indicates the close significance. However, for the stickiness result, 0.63, 0.55, 0.71, and 1.03 were classified as A, but why 0.71 and 0.76 were classified as B?

3.      On page 9, formulation D is described as having lower viscosity but showing higher stickiness from sensory evaluation. What’s the reason for this discrepancy? This needs to be explained, otherwise, there will be a question on the reliability of one of the results.

4.      Figure 4 should be called Table xx.

5.      On page 9, the author stated “This study confirms the positive correlation between shine, residue, and oiliness with surface tension”. However, the results of surface tension were not presented or discussed in the context.

6.      On page 12, the author concluded “formulation E (dimethicone 350) demonstrated superior skin protection against transepidermal water loss (TEWL).” Does this align with the “Dry Touch” in Figure 2? How to grade the “Dry Touch”; is the lower valve indicating less dryness? Were “Dry Touch” evaluated at time point T0 or T1 or T2 during sensory evaluation? All these experimental design details need to be elaborated.

7.      Pages 13 – 16: authors have over 4 pages on the packaging study through Neurosensory Analysis. There were lots of work involved, but unfortunately, reader is disappointed at the inclusion of this part in a scientific publication. This study investigates participant’s attention on different packaging or labels, which belongs to a marketing project instead of pharmaceutics research.  

8.      Page 16 Section 3.5.1 illustrated Consumer Profile and Product Preference based on the survey of 105 participants. However, in reality, there are already many plant-based products on the market so real consumption data will be more convincing to show customer’s willingness towards plant-based products. Unless the survey is based on the customer’s preference to the formulations designed by the authors in this manuscript, after their usage. Otherwise, such information is more suitable as background introduction of the manuscript.

9.      On page 18, consumers are questioning the sustainability and biodegradability of plant-based products. So how’s the biodegradability of policitronellol as compared to dimethicone? This is more of researchers’ responsibility to address the concern.

10.  The abstract is too lengthy, which would be better to be condensed and also avoid using “Objectives”, “Methods”, “Results” as linking words.

11.  The whole manuscript is too lengthy, especially with section 4 Discussion duplicating the results and conclusions in sections 3 and 5. Sections 3 and 4 can be combined in some way.

12.  Typo in the Abstract: “This study is na evaluation…”

Author Response

Dear Reviewer 3,

The responses to your valuable comments are included in the attached document.

Thank you very much.

Round 2

Reviewer 1 Report

Comments and Suggestions for Authors

I attached my comments. 

Author Response

Comments :

Unfortunately, the new version of the manuscript was not improved. There are some errors in interpreting the experimental data. The authors claimed that their emulsions exhibit thixotropic behavior, i.e. the viscosity is lowering as the time of shearing increases. I attached here a typical thixotropic structural breakdown behavior curve (from https://wiki.anton-paar.com/en/basics-of-thixotropy/).

As can be observed in figure 2 from the manuscript, the down curve is above the up curve, which is not a thixotropic behavior, does not suggest a structural breakdown, and a decrease of viscosity in time.

The meaning of the parameters from Table 2 was added, but the authors do not specify how they evaluated these parameters. Probably the experimental data were fitted using the well-known equation Herschel–Bulkley.

The authors do not explain how they calculated the thixotropy. This is not enough: “Thixotropy is expressed in Pa/s and was reported in the table as the mean values calculated from these tests, along with their respective standard deviations.” What tests???

In my opinion, the authors should remove the rheological analysis because their conclusions are not sustained by their experimental data and they could refer only to sensorial evaluation.

Answers:

Thank you for your comment. The error was not in the interpretation of the data, but in the caption of the curves in Figure 2. All the samples really showed thixotropic behavior, with the decrease of viscosity with time, as shown in the following graph. The methodology for calculating the thixotropy and determining the rheological parameters were added to the “Material and methods” section.

As rheological properties were not the main focus of this paper, it was summarized to avoid misunderstanding, showing only the results that can be related to the sensory evaluation.

Reviewer 2 Report

Comments and Suggestions for Authors

Variances meet the assumptions, although the criteria for normality of distribution, such as the results of the Shapiro-Wilk test, were not provided. The results of Levene's test indicate homogeneity of variances. However, the F-value was not greater than 1 in all cases, but the descriptions below the tables are correct.

In the measurement results, you stated that 10 measurements were performed, as recommended by the device manufacturer. However, it is generally recommended to conduct 3 repetitions at different points on the skin. Why was this procedure deviated from? The results obtained in this way also explain the relatively uniform error bars.

The error bars have been corrected, and the significance has been marked.

The authors addressed my comments. After minor clarifications, the work may be considered for publication.

Author Response

Reviewer 2

Comments :

Variances meet the assumptions, although the criteria for normality of distribution, such as the results of the Shapiro-Wilk test, were not provided. The results of Levene's test indicate homogeneity of variances. However, the F-value was not greater than 1 in all cases, but the descriptions below the tables are correct.

In the measurement results, you stated that 10 measurements were performed, as recommended by the device manufacturer. However, it is generally recommended to conduct 3 repetitions at different points on the skin. Why was this procedure deviated from? The results obtained in this way also explain the relatively uniform error bars.

The error bars have been corrected, and the significance has been marked.

The authors addressed my comments. After minor clarifications, the work may be considered for publication.

Answers:

Thank you for your comment. Below are the normality tests (Shapiro-Wilk) at the 1% significance level. p-values in bold were significant, indicating that the distribution does not follow normality in these cases. Only a few p-values were significant.

The vast majority of attributes exhibited normality. Additionally, we did not want to present two different types of tests in the same evaluation, so we maintained parametric tests for all attributes. Furthermore, the white residue attribute consistently shows results very close to zero, and any slightly higher assessment may already lead to non-normality. However, this does not affect the results of the comparison between samples, which had means of 0.01 (almost zero for all samples), and the outcome indicated no significant difference between them.

Summary of the Normality Tests for All Dependent Variables for Formulations A, B and C:

Dependent Variable

p-value

Spreadability

0.097

Slipperiness

0.733

Stickiness

0.394

Immediate White Residue

<0.0001 **

Residual White Residue

0.034 **

Velvety Film

0.437

Dry Touch

0.745

Immediate Shine

0.185

Residual Shine

0.182

Immediate Oiliness

0.874

Residual Oiliness

<0.0001 **

Immediate Greasy Film

0.770

Residual Greasy Film

0.118

Absorption Point

0.274

Summary of the Normality Tests for All Dependent Variables for Formulations A, D and E:

Dependent Variable

p-value

Spreadability

0.369

Slipperiness

0.000

Stickiness

0.512

Immediate White Residue

<0.0001

Residual White Residue

0.000

Velvety Film

0.766

Dry Touch

0.273

Immediate Shine

0.822

Residual Shine

0.430

Immediate Oiliness

0.862

Residual Oiliness

0.000

Immediate Greasy Film

0.095

Residual Greasy Film

0.335

Absorption Point

0.618

Summary of the Normality Tests for All Dependent Variables for Formulations A, F and G:

Dependent Variable

p-value

Spreadability

0.019

Slipperiness

0.842

Stickiness

0.677

Immediate White Residue

0.066

Residual White Residue

<0.0001

Velvety Film

0.170

Dry Touch

0.987

Immediate Shine

0.354

Residual Shine

0.454

Immediate Oiliness

0.880

Residual Oiliness

<0.0001

Immediate Greasy Film

0.207

Residual Greasy Film

<0.0001

Absorption Point

0.296

In the measurement results, you stated that 10 measurements were performed, as recommended by the device manufacturer. However, it is generally recommended to conduct 3 repetitions at different points on the skin. Why was this procedure deviated from?

The decision to perform 10 measurements, as recommended by the device manufacturer, was based on ensuring higher accuracy and reproducibility of the hydration measurements. While some studies suggest performing 3 repetitions at different points on the skin to account for regional variations, multiple measurements at the same site can reduce variability due to external factors such as ambient conditions, pressure applied to the probe, and slight skin surface irregularities.

Scientific studies on Corneometer measurement protocols indicate that increasing the number of repetitions can improve the reliability of results by minimizing outliers and measurement fluctuations. Furthermore, standardization protocols recommend adapting the number of measurements depending on the study design and required precision (Fluhr et al., 2006).

Appas, Catherine Tolomei Fabbron [UNIFESP]. A Influência De ésteres Emolientes Na Percepção Do Consumidor E Nas Características Sensoriais, Físico-químicas E Biometrológicas Quando Veiculados Em

Reviewer 3 Report

Comments and Suggestions for Authors

The authors have modified the manuscript to better support the statements by providing more data and correcting the misalignments. Regarding the scientific soundness of the manuscript, it's hard to be rated above average because the authors concluded that there is no direct relationship between certain physical properties vs sensory analysis in the cover letter. This makes the work more of reporting what was done instead of digging into the formulation property-ingredient relation and revealing general findings that can benefit a wide group of audience. The neurosensory analysis shifts the attention to the packaging instead of focusing on the formulation itself. It is not expected to fit in a pharmaceutics research paper, but it can be accepted to be an add-on to the manuscript.

Author Response

Reviewer 3

Comments :

The authors have modified the manuscript to better support the statements by providing more data and correcting the misalignments. Regarding the scientific soundness of the manuscript, it's hard to be rated above average because the authors concluded that there is no direct relationship between certain physical properties vs sensory analysis in the cover letter. This makes the work more of reporting what was done instead of digging into the formulation property-ingredient relation and revealing general findings that can benefit a wide group of audience. The neurosensory analysis shifts the attention to the packaging instead of focusing on the formulation itself. It is not expected to fit in a pharmaceutics research paper, but it can be accepted to be an add-on to the manuscript.

Answers:

Thank you for your valuable feedback and constructive insights. We appreciate your thorough evaluation of our manuscript and the points raised regarding the scientific depth and focus of our study.

First, we want to point out the importance and timely relevance of this study. There is a major move globally to more sustainable products, ingredients and manufacturing processes. This move is very well developed in consumer discretionary products and cosmetics and is being driven primarily by the consumer. Our study clearly demonstrates how the consumer or end user of the product evaluates the product based on a range of factors including the visual presentation presented. In the area of products applied to the skin, the cosmetic products and personal care products industries often lead the innovation in these ingredients, but their application of innovative ingredients and processes also informs the development of pharmaceutical products. As such, we believe this research is relevant and information for the readership of Pharmaceutics.

The appropriate selection of emollients plays a fundamental role in the development of topical formulations, directly influencing their rheological and sensory properties. Emollients affect the viscosity, spreadability, and absorption time of the product, impacting both physicochemical stability and consumer perception. The formulation's rheology, determined by the interaction between the oil phase, emulsifiers, and structuring polymers, modulates flow resistance and tactile sensation during application. Additionally, the choice of emollients can alter essential sensory attributes such as residual oiliness, perceived hydration, and drying time on the skin. Therefore, understanding the relationship between the formulation’s physical properties and its sensory acceptance is essential to optimize product performance and meet consumer expectations.

Recent studies have been conducted to establish the relationship between textural attributes and structural and physical characteristics. However, sensory analyses are time-consuming and require a well-trained panel of evaluators. Understanding the relationship between the physicochemical properties of an emollient and its sensory performance can significantly facilitate the work of formulators.

Viscosity is a well-established physicochemical characteristic in the literature regarding its correlation with the sensory attributes of spreadability and stickiness. Gorcea Mihaela and Donaa Laura (2013) observed in their studies that the higher the viscosity, the lower the spreadability, concluding that spreadability results depend on molecular weight, viscosity, and the chemical structure of the components (Gorcea Mihaela & Donaa Laura, 2013). In the study published by Parente et al. (2010), the authors identified a correlation between viscosity, difficulty in spreading, and stickiness. Additionally, shine, residue, and oiliness showed a positive correlation with surface tension (Parente et al., 2010).

When comparing formulations B and C, formulation B (polycitronellol HA) exhibited higher thixotropy and viscosity values than formulation C (dimethicone 100), resulting in lower spreadability and slip averages in the sensory analysis.

In the comparison between formulations D (polycitronellol H) and E (dimethicone 350), rheological results indicate that formulation D has lower viscosity than formulation E. However, formulation D showed higher averages and a significant difference only in the stickiness attribute, without significant variations in spreadability and slip.

Formulation F (polycitronellol compound) demonstrated a more pronounced pseudoplastic behavior, while formulation G (solid petrolatum) exhibited lower pseudoplasticity. Formulation G also presented the lowest viscosity and the highest thixotropy values. Additionally, this formulation exhibited a stress overshoot at low shear rates, indicating a more structured and resilient network—a behavior not observed in the other formulations.

Attributes such as spreadability, slip, immediate and residual white residue, immediate and residual oiliness, and immediate shine showed no significant differences among the formulations. However, the stickiness of formulation G was lower than that of formulation F.

Regarding the neurosensory analysis, our intention was to offer additional insights that could complement the overall evaluation of the formulation. This was specifically to evaluate the visual perception of the overall product and packaging and how that relates to their acceptance of the product. We acknowledge that this is of greater importance in cosmetic and personal care products that are a discretionary purchase. However, where a consumer has the option to choose from a range of pharmaceutical products with similar therapeutic efficacy, the environmental credentials then have a role to play in their product choice. This is particularly the case in “over the counter” topical pharmaceuticals where it is often easier for the consumer to choose their product. We would suggest that it may also be a factor in the choices of products that health practitioners recommend if they too consider environmental factors.

REFERENCE

Gorcea, M., & Laura, D. (2013, May 31). Evaluating the physiochemical properties of emollient esters for cosmetic use. Cosmetics & Toiletries. https://www.cosmeticsandtoiletries.com/testing/sensory/article/21836556/evaluating-the-physiochemical-properties-of-emollient-esters-for-cosmetic-use

Parente, M. E., Ares, G., & Manzoni, A. V. (2010). Application of two consumer profiling techniques to cosmetic emulsions. Journal of Sensory Studies, 25(5), 685–705. https://doi.org/10.1111/j.1745-459X.2010.00297.x

Round 3

Reviewer 1 Report

Comments and Suggestions for Authors

The authors reviwed the manuscript according to suggestions made.

Reviewer 2 Report

Comments and Suggestions for Authors

The authors revised the manuscript according to the comments. In my opinion, the manuscript can be considered for publication.